# TIR1/AFB-Aux/IAA auxin perception mediates rapid cell wall acidification and growth of Arabidopsis hypocotyls

Matyáš Fendrych[1], Jeffrey Leung[2], Jiří Friml[1]*

[1]Institute of Science and Technology Austria, Klosterneuburg, Austria; [2]Institut Jean-Pierre Bourgin, UMR1318 INRA-AgroParisTech, INRA - Centre de Versailles-Grignon, Saclay Plant Science, Versailles, France

**Abstract** Despite being composed of immobile cells, plants reorient along directional stimuli. The hormone auxin is redistributed in stimulated organs leading to differential growth and bending. Auxin application triggers rapid cell wall acidification and elongation of aerial organs of plants, but the molecular players mediating these effects are still controversial. Here we use genetically-encoded pH and auxin signaling sensors, pharmacological and genetic manipulations available for Arabidopsis etiolated hypocotyls to clarify how auxin is perceived and the downstream growth executed. We show that auxin-induced acidification occurs by local activation of $H^+$-ATPases, which in the context of gravity response is restricted to the lower organ side. This auxin-stimulated acidification and growth require TIR1/AFB-Aux/IAA nuclear auxin perception. In addition, auxin-induced gene transcription and specifically SAUR proteins are crucial downstream mediators of this growth. Our study provides strong experimental support for the acid growth theory and clarified the contribution of the upstream auxin perception mechanisms.

*For correspondence: jiri.friml@ist.ac.at

**Competing interests:** The authors declare that no competing interests exist.

## Introduction

Plants support their bodies by a hydrostatic skeleton that consists of pressurized cells encased by a strong extracellular composite, the cell wall (*Höfte, 2014*). This has several crucial consequences for plant multicellularity: (vascular) plant cells do not move, and they grow mostly symplastically (no sliding occurs; reviewed in *Lintilhac, 2014*). Thus plant shapes and patterns are sculptured by the precise orientation of cell division and tight control of growth. Growth depends on a balance between the turgor pressure (*Ray et al., 1972*) and the yielding of the cell wall to this pressure (*Heyn, 1940*; *Lockhart, 1965*).

The phytohormone auxin represents an excellent case study of growth control. Auxin induces rapid (minutes to hours) cell elongation in the aerial organs. In fact, this feature was used to define and discover auxins in a coleoptile bending test (e.g. *Went and Thimann, 1937*). More specifically, applied auxin induces growth of decapitated organs that are depleted of auxin. Auxin-induced growth is characterized by acidification of the cell wall. Acidification of the growth media is sufficient to trigger growth in certain conditions; therefore the term 'acid growth' is often used (reviewed in *Kutschera, 1994*). Auxin induces extrusion of protons into the apoplast, activating expansins in the cell wall (*McQueen-Mason, 1992*), which in turn leads to weakening of the cell wall and to growth. The central players in this process are the plasma membrane (PM) P-type $H^+$-ATPases – AHAs. By inducing the expression of SAUR19 protein that inactivates a PP2C-family phosphatase, auxin application results into phosphorylation of the critical Thr residue in the autoinhibitory domain of AHA, which leads to activation of the proton pump (*Takahashi et al., 2012*; *Spartz et al., 2014*). At the same time, by pumping protons into the apoplastic space, the PM is hyperpolarized, which drives

the opening of voltage-sensitive $K^+$ inward-rectifying channels (*Hedrich et al., 1995*), leading to accumulation of $K^+$ ions and turgor increase in the cell. On the other hand, the acid growth theory does not seem to be valid in the roots (*Pacheco-Villalobos et al., 2016*).

Auxin-induced growth experiments are relevant for the growth and development of plants, and have a direct parallel in the movement of plant organs during tropisms – directional movements. According to the Cholodny-Went theory, the asymmetrical distribution of auxin leads to the bending of the organ (*Went and Thimann, 1937*). Upon a directional stimulus of the shoot, auxin is redistributed by the PIN and ABCB proteins to the cells of the side which will elongate (*Friml et al., 2002*; *Noh et al., 2001*; *Rakusová et al., 2011*). Consequently, these cells experience a sudden increase of auxin concentration and react by elongation. Also more generally, auxin is crucial for growth of aerial organs, as stated by *Went and Thimann (1937)*: "Ohne Wuchsstoff kein Wachstum – without auxin no growth".

Currently, the literature considers at least two receptors for auxin: the nuclear TIR1/AFB Aux/IAA co-receptor (*Dharmasiri et al., 2005*; *Kepinski and Leyser, 2005*), and the Auxin Binding Protein 1 (ABP1) (*Löbler and Klämbt, 1985*; reviewed in *Grones and Friml, 2015*). The TIR1 pathway operates by the de-repression of transcription of auxin-induced genes, while the ABP1 pathway was proposed to regulate activities of proteins directly, and thus can operate immediately after auxin perception. Although auxin-induced growth is one of the oldest experimental questions in plant physiology, there is no agreement on how auxin is perceived during this process. This controversy stems from several facts: auxin-induced growth starts only approximately 15 min after application (*Kutschera, 1994*), and requires novel protein synthesis (*Edelmann and Schopfer, 1989*), hinting to the involvement of the nuclear TIR1/AFB Aux/IAA pathway. The major support for the nuclear auxin pathway involvement is the regulation of AHA activity by SAUR19 (*Spartz et al., 2014*), a member of a family of very small proteins the expression of which is rapidly induced by auxin. Recently, the significance of the ABP1 pathway has been undermined by the lack of any obvious developmental phenotypes of the *abp1* knockout mutants (*Grones et al., 2015*; *Gao et al., 2015*; *Michalko et al., 2015*, *2016*). Despite these recent findings, there is also evidence demonstrating the importance of the ABP1 pathway, obtained by approaches and methods other than knock-out mutants. In protoplasts auxin induces rapid PM hyperpolarization and swelling that can be blocked by ABP1 antibodies (*Leblanc et al., 1999*; *Yamagami et al., 2004*). Moreover, *Schenck et al. (2010)* concluded that the TIR1/AFB pathway is not involved in auxin-induced elongation, based on the fact that a quadruple *tir1/afb* mutant still responds to auxin by elongation. Furthermore, *Takahashi et al. (2012)* report that the AHA PM $H^+$-ATPase is still phosphorylated, and thus activated, upon auxin application in a double *tir1-1 afb2-3* mutant.

In the current literature the question how is auxin perceived and what downstream processes are involved in auxin-induced growth remains unanswered (e.g. *Dünser and Kleine-Vehn, 2015*). Here we analyze the components of the response – auxin signaling, cell wall acidification and growth. We aim to solve how auxin is perceived during auxin-induced growth, and whether there are any rapid, 'non-genomic' components of this response. By analyzing lines that overexpress stabilized version of the SAUR protein or by genetically activating the PM $H^+$-ATPases we attempt to clarify whether the acid growth hypothesis is valid in the Arabidopsis hypocotyl. Finally, we test how the results from the auxin-induced growth experiments relate to the *in vivo* situation on the example of gravitropic response. The combination of genetic tools and techniques that have become available enabled us to systematically re-evaluate this classical question in plant biology.

## Results and discussion

### Auxin-induced growth, acidification, and transcriptional response are tightly correlated

When grown in darkness, hypocotyls of plants rapidly elongate, and this elongation is fueled by the auxin from the cotyledons. Therefore, the etiolated hypocotyl is an excellent experimental model for auxin-induced elongation (*Schenck et al., 2010*; *Takahashi et al., 2012*). In this work we used three-day-old etiolated hypocotyls of *Arabidopsis thaliana*. We used the auxin analog NAA, to ensure that auxin penetrates the hypocotyl segment well, and is not degraded by light during the experiments. NAA is stable, readily enters cells by diffusion, and is a substrate for auxin efflux carriers

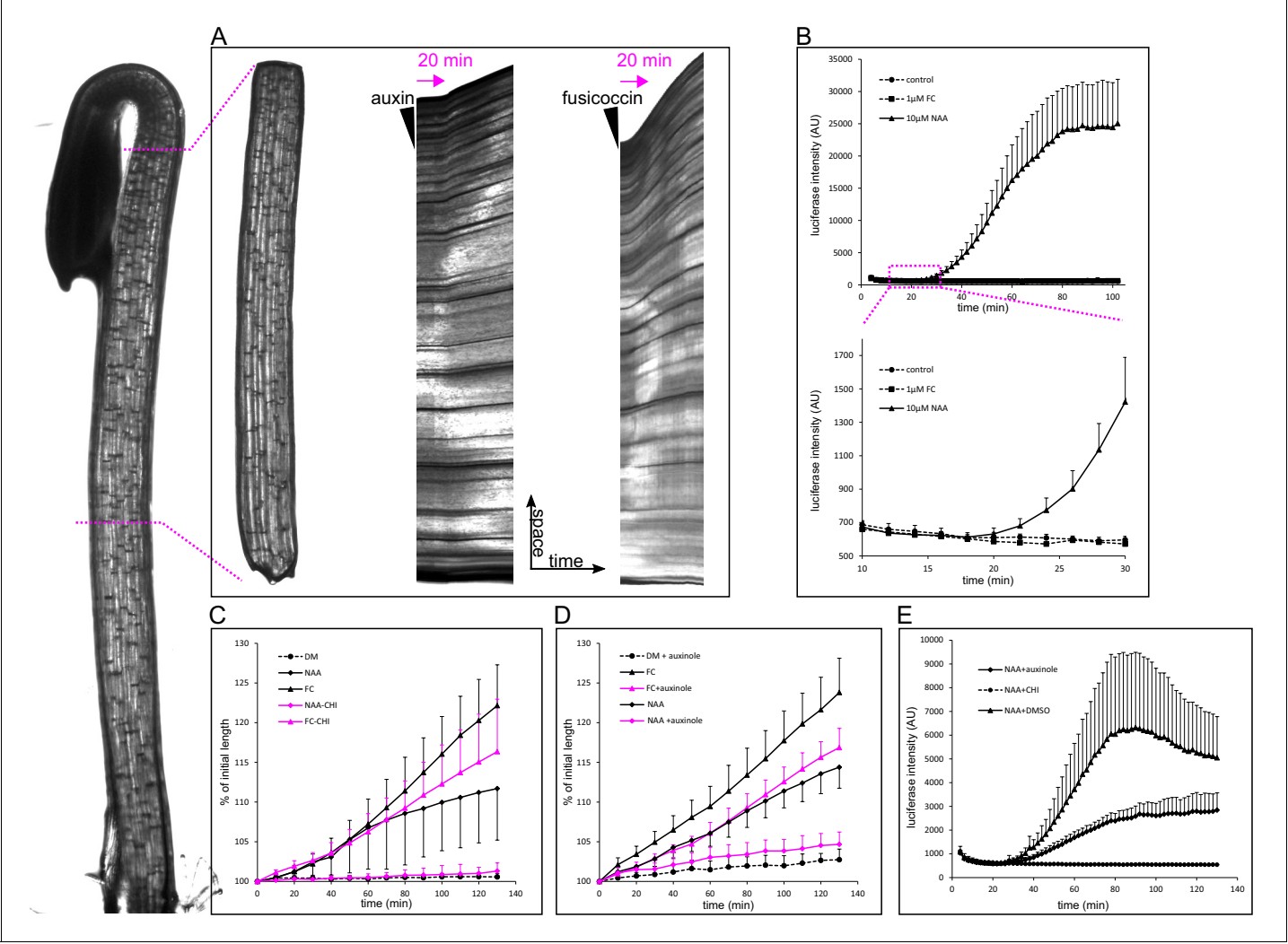

**Figure 1.** Auxin-triggered growth of hypocotyl segments correlates with nuclear auxin signaling. (**A**) A brightfield image of a three-day old etiolated Arabidopsis hypocotyl and the corresponding decapitated hypocotyl segment used in the elongation assay. On the right, kymographs of growing hypocotyls right after application of 10 µM NAA or 5 µM FC (black arrowhead). A lag phase of 20 min preceding the rapid growth is apparent in the case of NAA. The lag phase is shorter in the case of FC. Arrow indicates 20 min. (**B**) Transcriptional auxin response of hypocotyl segments visualized by the DR5::LUC reporter. Auxin response can be detected 20 min after auxin application. There is no response in the control and fusicoccin treatment. Each curve corresponds to 4 hypocotyl segments, errorbars are stdev.s. The lower graph depicts the onset of DR5::LUC response. (**C**) Hypocotyl segments do not elongate on the depletion medium (DM); auxin (10 µM NAA) and 1 µM fusicoccin (FC) trigger segment elongation. The protein synthesis inhibitor cycloheximide (50 µM CHI) completely blocks auxin-induced elongation, while FC-induced elongation is only mildly affected by CHI. (**D**) Auxinole (50 µM), an inhibitor of the TIR1/AFB auxin receptor, largely inhibits auxin induced hypocotyl elongation; the FC-triggered elongation is less affected. In (**C**) and (**D**), each curve corresponds to 6 hypocotyl segments, auxinole and CHI were applied during the depletion phase (30 min before auxin treatment); errorbars are stdev.s. (**E**) CHI completely blocks DR5::LUC auxin response, and auxinole diminishes the response. NAA (10 µM) was applied at timepoint 0; auxinole and CHI (both 50 µM) were present during the depletion phase (30 min before auxin treatment). Quantification as in **B**).

The following source data is available for figure 1:

**Source data 1.** Numerical data for the graphs.

(*Delbarre et al., 1996*). In the etiolated hypocotyl of *Arabidopsis thaliana*, depleted of endogenous auxin by decapitation, auxin induced growth with a delay of approximately 20 min (19.75 ± 2.9 min stdev, n = 12) (*Figure 1A* and *Figure 2A*). To ensure that this delay is not an artifact of the experimental system, we used the fungal toxin fusicoccin (FC; *Ballio et al., 1964*) that activates the PM H⁺-

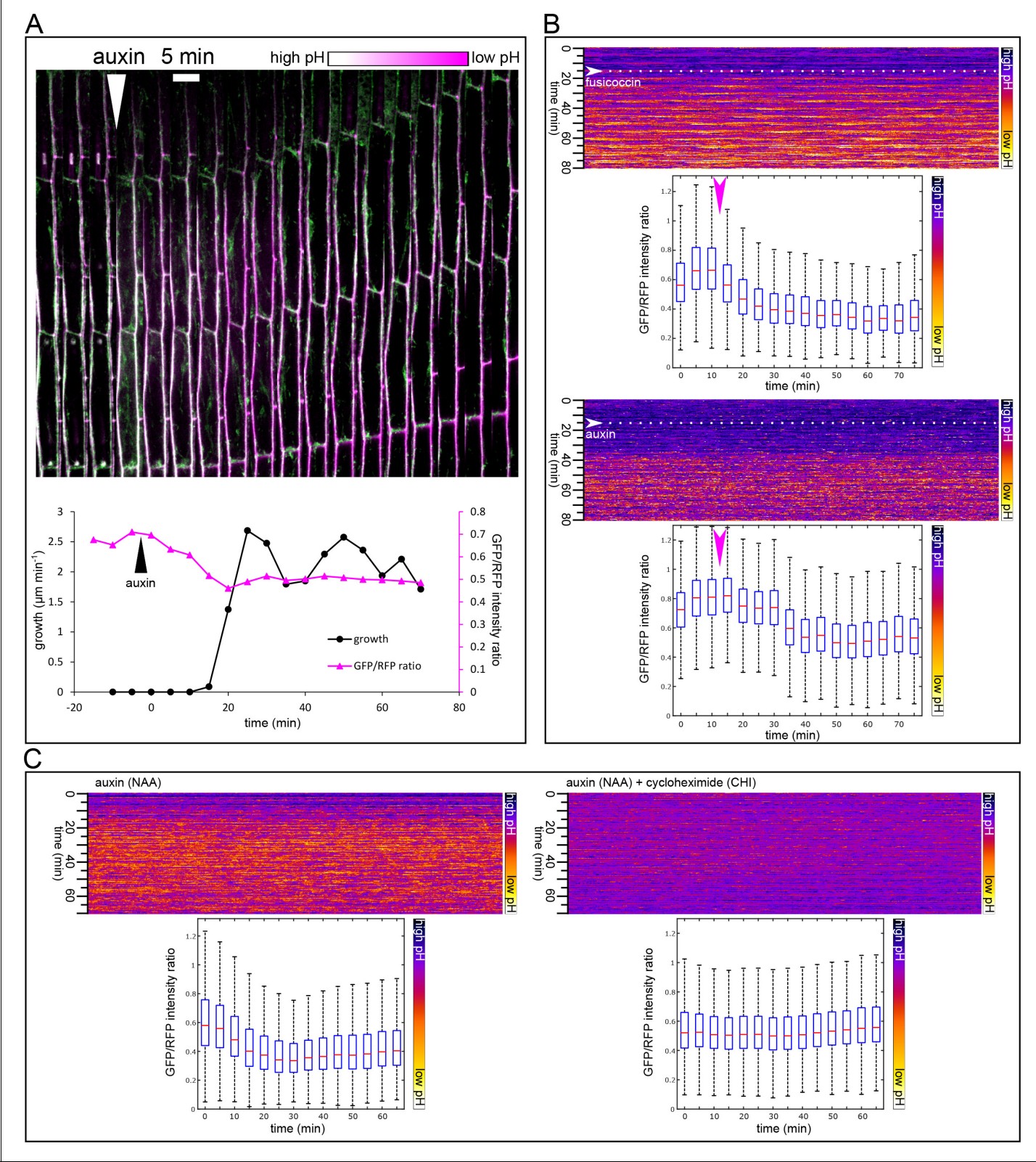

**Figure 2.** Apoplast acidification visualized by the apo-pHusion marker line. (**A**) Apoplast acidification and cell elongation follow approximately 20 min after auxin application (white arrowhead). A region of a hypocotyl segment of the apo-pHusion marker line is shown in time (timestep 5 min). Apoplastic pH is approximated by the ratio between RFP (insensitive to pH) and GFP intensities (quenched by low pH). RFP is shown in magenta, GFP in green; higher pH appears white and lower pH magenta. Bellow, a quantification of growth and apoplastic pH of the same hypocotyl segment. (**B**) *Figure 2 continued on next page*

*Figure 2 continued*

AreaKymo visualisation illustrates the apoplastic pH drop that occurs immediately after fusicoccin addition (5 µM), upper panel. In contrast, a lag phase of 20 min follows auxin application (10 µM NAA), after which apoplastic pH drops suddenly (lower panel). FC and NAA were added during imaging as indicated by arrowheads and dotted lines. Each timepoint is represented by a rectangle and consists of cell wall pixels of the original apo-pHusion confocal image; time progresses from top to bottom. The LUT key is shown on the right. Bellow a quantification of the pH drop shown as a series of boxplots. A time series imaging of a single hypocotyl segment was used as input data. (C) Auxin-induced apoplastic pH drop requires protein synthesis. An AreaKymo after application of auxin (10 µM NAA) 4.5 min before imaging. Cycloheximide (CHI 50 µM) completely blocks apoplastic acidification that is apparent in the control situation. Bellow, the quantification is shown; each plot consists of a confocal timelapse imaging of 4 hypocotyl segments.

The following source data is available for figure 2:

**Source data 1.** Numerical data for the graph 2A.

ATPase by enabling binding of an activating 14-3-3 protein to the pump (*Baunsgaard et al., 1998*), and so triggers apoplast acidification and growth. Application of FC triggered growth in 6–8 min after its application (*Figure 1A*). To monitor the transcriptional response to auxin, we needed a reporter that would be more rapid than transcriptional GFP reporters, as fluorescent proteins require significant time to mature and become fluorescent (*Shaner et al., 2005*). The Firefly luciferase enzyme is, on the other hand, active immediately after its translation (*Kolb et al., 1994*). Therefore we used the auxin response reporter DR5 driving the expression of Firefly luciferase enzyme – DR5:: LUC (*Moreno-Risueno et al., 2010*). Accordingly to the growth onset, nuclear auxin response could be detected 20 min after auxin application (*Figure 1B*). To visualize the auxin-induced acidification and thus approximate the activity of the PM $H^+$-ATPases, we used the genetically encoded apoplastic pH sensor apo-pHusion (*Gjetting et al., 2012*). Similar to the growth initiation and the nuclear auxin signaling, a sudden drop in apoplastic pH could be detected 20 min (20.11 ± 5 min stdev, n = 12) after auxin application (*Figure 2A*; *Video 1*). We developed a MATLAB-based tool that we named AreaKymo in order to be able to visualize and quantify apoplastic pH changes in an unbiased way. The apo-pHusion pH sensor consists of mRFP (insensitive to pH changes) and GFP (responsive to pH changes). AreaKymo selects the cell wall areas based on RFP fluorescence, measures RFP and GFP intensities in each pixel, and plots these pixels in a time-space plot. Using Area-Kymo, the delay between auxin application and apoplast acidification is clearly visible (*Figure 2B*). Again, in the case of FC, the apoplast acidification followed immediately after its application (*Figure 2B*, *Video 2*). The 'acid growth' literature often discusses the necessity to abrade the stem segments in order to achieve proper penetration of the drugs into the tissue (*Kutschera, 1994*). Our results show that in the case of the comparably minute Arabidopsis hypocotyl segments, auxin enters the tissue rapidly, inducing growth, acidification, and transcriptional auxin response within approximately 20 min. If we take into account the velocity of auxin transport in Arabidopsis ~8 mm hr$^{-1}$ (*Kramer et al., 2011*), auxin should reach the middle of the hypocotyl segment (~1.2 mm) in ~4.5 min. Our results are thus in a good agreement with the published timescale of auxin-induced elongation (17 min in *Kutschera, 1994*). Most importantly, we directly visualized auxin-induced acidification, transcriptional response to auxin, and segment elongation, all of which are tightly correlated in time and happen with an approximately 20-min lag phase after auxin application.

## Auxin-induced growth requires de novo protein synthesis

To test whether protein synthesis is necessary for these rapid auxin responses, we pre-treated hypocotyl segments with cycloheximide (CHI), a protein synthesis inhibitor. As a control to test for the ability of the cells to grow, we used FC, the application of which triggers growth, bypassing the entire auxin signaling pathway (*Figure 1B,C*). Application of CHI completely blocked auxin-induced growth (*Figure 1C*), while FC-induced growth was only slightly decreased. This decrease was evident in the later phases of growth, when the lack of protein synthesis might affect the overall status of the cell (*Figure 1C*). This result demonstrates that protein synthesis is not necessary for growth as such; hypocotyl segments are able to grow for some time without protein synthesis. However, protein synthesis is absolutely required for *auxin*-induced growth. Furthermore, CHI treatment completely inhibited the auxin-induced apoplast acidification (*Figure 2C*), and auxin response, visualized by the

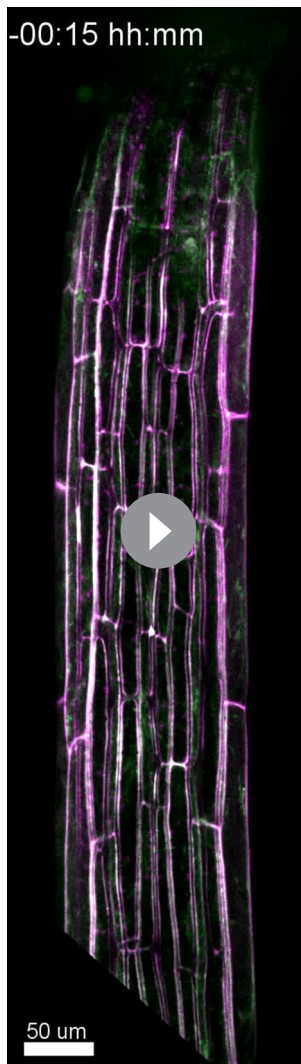

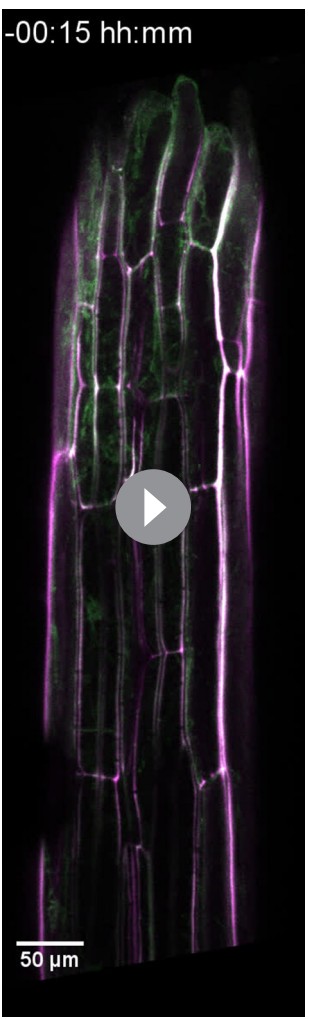

**Video 1.** The movie shows an apo-pHusion marker line hypocotyl segment.	NAA (10 µM) is added at time 0, acidification follows in approximately 20 min. The pH-sensitive GFP is shown in green, the reference mRFP in magenta, therefore white color corresponds to higher pH, while magenta to lower pH values.

**Video 2.** Apoplastic pH drops rapidly after adding fusicoccin (5 µM at timepoint 0).	An apo-pHusion marker line hypocotyl segment. The pH-sensitive GFP is shown in green, the reference mRFP in magenta, therefore white color corresponds to higher pH, while magenta to lower pH values.

DR5::LUC reporter (*Figure 1E*). This result is consistent with previous results in maize coleoptiles and pea internode segments, where CHI blocked auxin-induced growth (*Lado et al., 1977*; *Edelmann and Schopfer, 1989*; *Kutschera, 1994*). Other studies reported auxin-induced growth in the presence of CHI (*Rose, 1974*). It was also shown that Small Auxin Up mRNAs (SAUR) accumulate in hypocotyl segments after CHI treatment (*Franco et al., 1990*), but without translation, these mRNAs are not able to trigger growth. In our experimental setup, the total inhibition of the DR5:: LUC response by CHI leaves no space for doubts that the CHI treatment was effective, and so demonstrates the necessity of protein synthesis for the auxin growth response, further supporting the involvement of the nuclear auxin pathway. Finally, we used auxinole that binds to the TIR1 auxin co-receptor and blocks the formation of the TIR1-auxin-Aux/IAA complex (*Hayashi et al., 2012*). Treatment with auxinole led to a decrease in the nuclear auxin response visualized by DR5::LUC (*Figure 1E*), and resulted in a large decrease in auxin-induced growth (*Figure 1D*), while FC-induced growth still occurred. This result strengthens the hypothesis that the nuclear auxin receptors are

perceiving auxin in the hypocotyls to mediate apoplastic pH changes, growth, and probably maintenance of the turgor pressure, necessary for continued growth.

## Auxin-induced growth requires TIR1/AFB-Aux/IAA signaling

Previous works established that multiple mutations in the TIR1/AFB pathway do not inhibit the auxin-induced growth of the hypocotyl (*Takahashi et al., 2012*; *Schenck et al., 2010*), and it was suggested that auxin is perceived by the ABP1 pathway, the null mutants of which were not available at that time. We therefore set out to test the new mutants in the ABP1 gene (*Gao et al., 2015*). The new *abp1* null mutants' reaction to auxin was identical to that of the control (*Figure 3A*). In the works using the multiple *tir1/afb* mutants, the nuclear auxin pathway was not completely eliminated, most likely due to the redundancy of the 6 TIR1/AFB receptors. It is also important to note that the development of the higher order mutant is seriously affected, and the phenotypes range from normal-looking plants to seriously stunted plants lacking a root (*Dharmasiri et al., 2005*), making it very

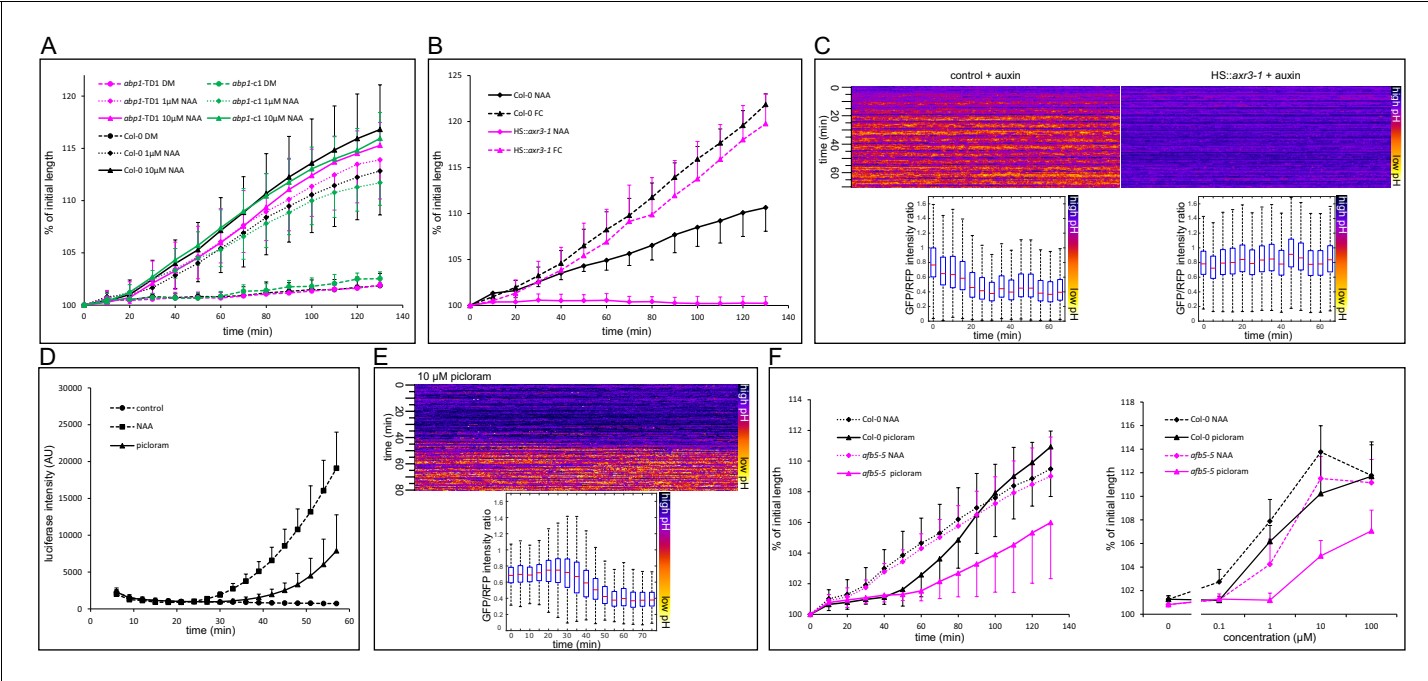

**Figure 3.** Nuclear auxin signaling is needed for growth and apoplast acidification. (**A**) Mutations in the ABP1 gene do not influence auxin-induced hypocotyl elongation. (**B**) Inhibition of the TIR1/AFB Aux/IAA auxin receptor by the dominant-negative *axr3-1* protein completely blocks auxin-induced (10 μM NAA) hypocotyl elongation. The elongation triggered by fusicoccin (1 μM FC) is not affected in the same line. Both *HS::axr3-1* and Col-0 seedlings were heatshocked 2 hr before auxin application. (**C**) Induction of the *HS::axr3-1* line completely blocks the auxin-induced apoplastic pH drop. AreaKymo shows that in the control (Col-0 x apo-pHusion), apoplastic pH drops after addition of auxin (10 μM NAA), while this drop is absent in the *HS::axr3-1* background. Each plot was constructed from 4 hypocotyl segments, both lines were heat-shocked 2 hr before imaging. (**D**) The auxin analogue, picloram (10 μM) triggers DR5::LUC auxin response; the response is slower than that of NAA (10 μM). Each curve corresponds to 4 hypocotyl segments, errorbars are stdev.s. (**E**) Picloram (10 μM) was applied before imaging, apoplastic pH drop occurs approximately 40 min later. An AreaKymo based on 3 hypocotyl segments; quantification is shown in the lower part of the panel. (**F**) A mutation in the AFB5 auxin co-receptor, *afb5-5*, leads to decreased sensitivity to picloram-induced hypocotyl elongation. On the left, a timecourse of elongation after 10 μM NAA or picloram is shown. On the right, a dose response curve of Col-0 and *afb5-5* to NAA or picloram is presented. In A,B,F, curves correspond to 6 hypocotyl segments, errorbars represent stdev.s.

The following source data and figure supplements are available for figure 3:

**Source data 1.** Numerical data for the graphs.

**Figure supplement 1.** Auxin induced growth of the *tir1/afb2/afb3* mutant and the non-induced *HS::axr3-1* line.

**Figure supplement 1—source data 1.** Numerical data for the graphs.

difficult to assess their physiology. As reported previously, a *tir1-1/afb2-1/afb3-1* triple mutant still responded normally to auxin (*Figure 3—figure supplement 3A*). In our hands, the quadruple *tir1-1/afb1-3/afb2-3/afb3-4* mutant was hardly able to form a hypocotyl in which we could assess the growth reaction to auxin. To circumvent the problem with the redundancy in the TIR1/AFB, we focused on the second part of auxin co-receptor – the Aux/IAA proteins. We used the *axr3-1* mutant that harbors a mutation in the DII domain of the IAA17 gene, leading to a semi-dominant auxin-resistant phenotype (*Leyser et al., 1996*), expressed from the heat shock-inducible promoter – HS::axr3-1 (*Knox et al., 2003*). After induction of the *axr3-1* protein, hypocotyls responded to FC treatment by rapid growth identical to that of control, but their reaction to auxin was completely inhibited (*Figure 3B*). Without induction, the behavior was identical to that of the control (*Figure 3—figure supplement 3B*). We introduced the apoplastic pH sensor apo-pHusion into the HS::axr3-1 background and analyzed the apoplastic acidification. Induction of the *axr3-1* protein completely prevented the auxin-induced apoplastic acidification (*Figure 3C*), fitting with the original agravitropic and short hypocotyl phenotype of *axr3-1* (*Leyser et al., 1996*). These results again show that auxin-induced growth is downstream of the TIR1/AFB Aux/IAA co-receptor. Even the auxin-induced apoplast acidification requires the degradation of Aux/IAAs and auxin-induced gene expression, and is not mediated directly by a putative auxin receptor.

To test the involvement of the TIR1/AFB pathway yet in another way, we exploited the fact that the AFB5 receptor is responsible for perception of the synthetic auxin picloram (*Walsh et al., 2006*; *Prigge et al., 2016*; *Calderón Villalobos et al., 2012*). In the Col-0 background, picloram induced growth, DR5::LUC auxin response, and apoplast acidification, although more slowly than NAA (*Figure 3D–F*). We then tested the growth response of the mutant in the AFB5 receptor, *afb5-5*. As expected, it was less sensitive to picloram (*Figure 3F*) in the auxin-induced hypocotyl elongation, while the reaction to NAA and fusicoccin was comparable to that of control (*Figures 3F* and *4C*). The auxin signaling pathway exhibits a homeostatic nature and contains several negative and positive feedback loops. For example, it is known that auxin-induced genes are able to suppress auxin response (*Mellor et al., 2016*). The combination of the *afb5-5* mutant, a normal looking plant, with the picloram auxin analogue treatment creates an unprecedented situation, where the decreased reaction of the *afb5-5* mutant to the auxin analogue picloram can get manifested and confirms the crucial role of the TIR1/AFB pathway in auxin-induced hypocotyl elongation. It remains a question why the picloram-induced DR5 response, acidification and growth show a delay when compared to NAA. It is possible that AFB5 has partially different functions than the other TIR/AFBs or that picloram needs more time to enter the cells of the hypocotyl segment.

## Stabilization of SAUR19 protein leads to auxin-independent growth

We have demonstrated by analysis of the processes, by pharmacological manipulations, and using genetic tools that auxin-induced growth and acidification is downstream of the nuclear TIR1/AFB Aux/IAA co-receptor. To address the downstream components involved, we looked at the SAUR proteins. SAURs form a family of very small, short-lived proteins that are rapidly induced by auxin application (*Ren and Gray, 2015*), and act downstream of the TIR1/AFB Aux/IAA auxin co-receptor. In a recent publication, *Spartz et al. (2014)* showed that the SAUR19 protein indirectly activates the PM H$^+$-ATPase. They found that tagging SAUR19 with GFP leads to its stabilization and confers overexpression phenotypes. The *35S::GFP-SAUR19* line evoked up to 35% higher H$^+$-ATPase activities and increased cell expansion, as compared to Col-0. The *35S::GFP-SAUR19* line behavior in our experimental system was remarkable. After decapitation, the hypocotyl segments elongated on the control medium without auxin, and the addition of auxin did not lead to further stimulation of growth (*Figure 4A*). We analyzed the apoplastic pH in the F1 cross between the *35S::GFP-SAUR19* and the apo-pHusion marker line. In the control (Col-0 x apo-pHusion) the apoplastic pH rises and growth stops quickly after decapitation, while apoplastic pH remained acidic in the *35S::GFP-SAUR19* background and the growth continued (*Figure 4F*). SAUR genes are arranged in clusters on chromosomes and encode very small proteins, which likely enables very rapid production of these proteins upon perception of auxin. Moreover, SAUR proteins are extremely short-lived (*Ren and Gray, 2015*), meaning that transcriptional regulation is a crucial determinant of their abundance. Stabilization of SAUR proteins leads to increased cell elongation (*Spartz et al., 2012*; *2014*; *Chae et al., 2012*). These properties make SAUR proteins a suitable 'gearbox' of growth that is able to rapidly respond to changing conditions, while involving the relatively slow

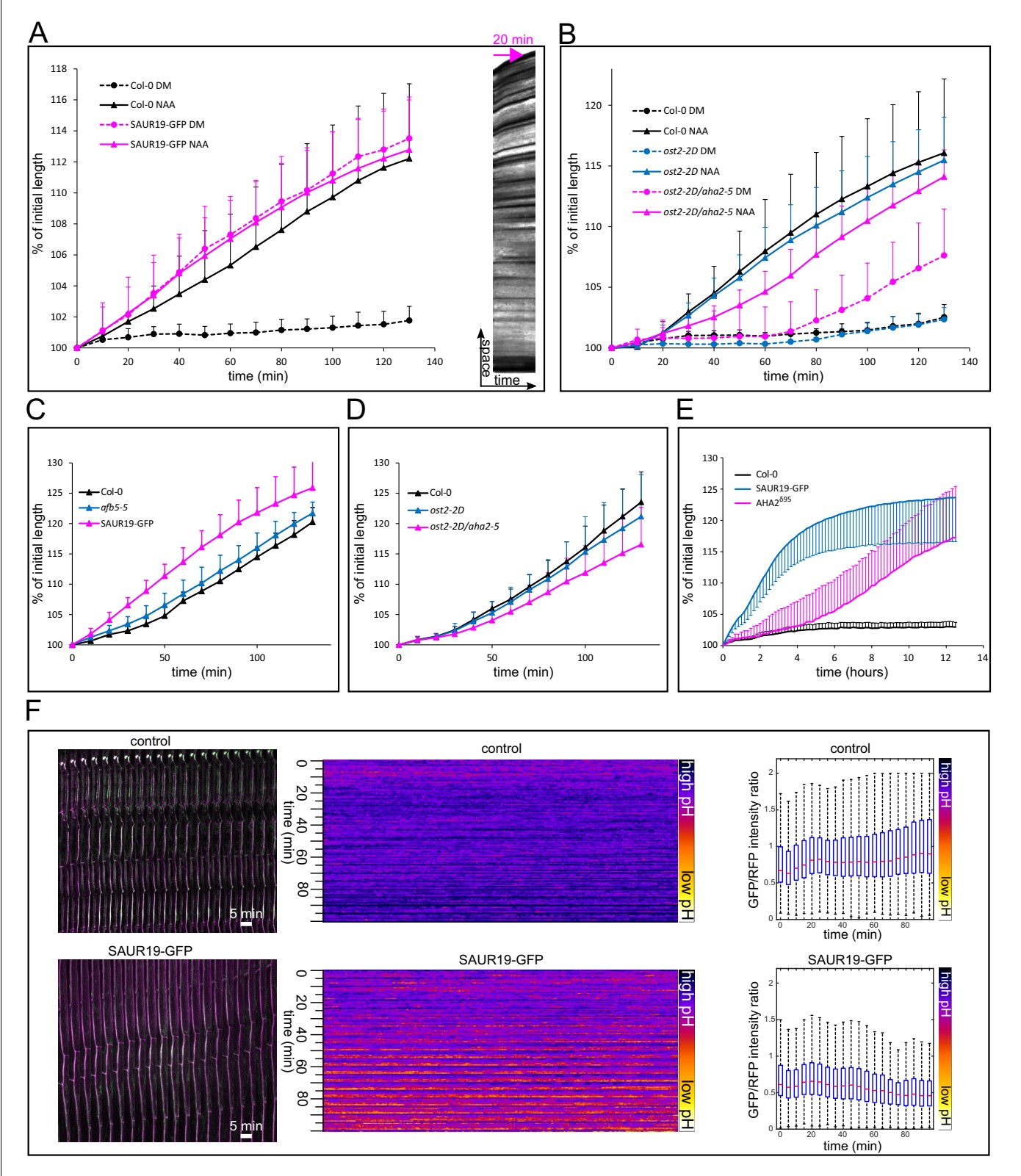

**Figure 4.** Activation of PM H⁺-ATPases triggers auxin-independent growth. (**A**) The *35S::GFP-SAUR19* hypocotyls elongate in the absence of auxin, and the addition of auxin (10 μM NAA) does not further increase their growth. On the right, a kymograph of a decapitated hypocotyl elongating in the absence of auxin is shown. Arrow represents 20 min. (**B**) The *ost2-2D* mutant line with a constitutively active AHA1 PM H⁺-ATPase does not cause an auxin-independent hypocotyl growth; while the *ost2-2D/aha2-5* mutant line elongates in the absence of auxin. (**C, D**) Elongation of *afb5-5*, *35S::GFP-*

*Figure 4 continued on next page*

*Figure 4 continued*

*SAUR19*, *ost2-2D* and *ost2-2D/aha2-5* in the presence of 1 µM fusicoccin. In A-D curves correspond to 6 hypocotyl segments, errorbars represent stdev.s. (E) Hypocotyl segments of $AHA2^{\delta95}$start to elongate approximately four hours after inducing the transgene expression. The autonomous elongation of *35S::GFP-SAUR19* is shown for comparison. Segments were placed on depletion medium with (Col-0 and $AHA2^{\delta95}$) or without (*35S::GFP-SAUR19* and another Col-0) 1 µM dexamethasone. Curves correspond to 7–10 hypocotyl segments, errorbars represent stdev.s. (F) *35S::GFP-SAUR19* line has an acidic apoplast and keeps elongating after decapitation in the absence of auxin, while the apoplastic pH of the control rises after decapitation and growth ceases. On the left, a region of a hypocotyl segment of the *35S::GFP-SAUR19* or Col-0 in the apo-pHusion background is shown in time (timestep 5 min). On the right, apoplastic pH of both lines is plotted and quantified using the AreaKymo tool.

The following source data is available for figure 4:

**Source data 1.** Numerical data for the graphs.

transcription and translation steps. Our results show that SAUR proteins mediate the signal between nuclear auxin perception and auxin-induced growth execution, but it is important to note that the conclusion is based mainly on the overexpression of a stabilized version of the SAUR19 protein.

## Activation of PM H⁺-ATPases is sufficient to trigger growth

The stabilized *35S::GFP-SAUR19* overexpression line supposedly stimulates growth by activation of the AHA PM H⁺-ATPases (*Spartz et al., 2014*), supporting the acid growth hypothesis for hypocotyls. In Arabidopsis, the AHA family has 11 members, of which AHA2 and its close homolog AHA1 together contribute about 70–80% of the total H⁺-ATPase activity. While individual *aha1* or *aha2* showed only modest phenotypes such as impaired root growth on alkaline pH, the double mutant is embryo-lethal (*Haruta et al., 2010*; *Haruta and Sussmann, 2012*). To clarify the functional relationship of auxin and H⁺-ATPases directly *in planta*, we took advantage of the availability of dominant mutations (*ost2*) available for the *AHA1* locus that render the pump constitutively active (*Merlot et al., 2007*). Although the over-expressing *35S::GFP-SAUR19* transgenic lines have been described to behave comparably to the *ost2-2D* mutant in several tests, we found that decapitated *ost2-2D* showed no or minor auxin-independent growth (*Figure 4B*). As constitutive mutations are not known for AHA2, we nonetheless created a double mutant by combining the *aha2-5* knock out with the *ost2-2D*, so that of the two main isoforms of AHAs, one is constitutively active and the other one absent. Indeed, the *ost2-2D/aha2-5* hypocotyl segments elongated in the absence of auxin (*Figure 4B*), more than the *ost2-2D* single mutants. The *ost2-2D/aha2-5* also showed a slightly decreased reaction to fusicoccin compared to *ost2-2D* and Col-0 (*Figure 4D*). These results provided additional insight into the functions of the two pump subunits: not only AHA1 and AHA2 play redundant roles, they are also interdependent. At least a notable fraction of the mature PM H⁺-ATPases in the plant cell might be hetero-oligomers of AHA1 and AHA2 subunits (a mature pump is thought to be a hexamer; *Kanczewska et al., 2005*). In the complete absence of AHA2, the equilibrium in the composition of the mature pump complex would shift towards more of the *ost2-2D* subunits. However, the ability of *ost2-2D/aha2-5* to elongate without auxin did not reach the magnitude of the *35S::GFP-SAUR19* plants. There are several possible explanations of this difference. Other of the 11 PM H⁺-ATPases might be activated by the SAUR proteins, or other sites of the auto-inhibitory loop might be regulated than those mutated in *ost2-2D*. Alternatively, the 'acid growth' explanation of SAUR action is not sufficient, and SAURs might target multiple proteins that initiate the growth response; and activating PM H⁺-ATPases is only a part of their action.

To distinguish between these possibilities, we used the line that inducibly expresses AHA2, devoid of its autoinhibitory domain – $AHA2^{\delta95}$ (*Pachecon-Villalobos et al., 2016*). Approximately 4 hr after transfer on the induction medium, hypocotyl segments of $AHA2^{\delta95}$started to elongate (*Figure 4E*). The auxin-independent growth of $AHA2^{\delta95}$was similar to that of *35S::GFP-SAUR19*. Based on these results we conclude that in the hypocotyls, strong activation of PM H⁺-ATPases is sufficient to trigger growth and can explain the auxin-independent elongation phenotype caused by the overexpression of stabilized SAUR19 protein. The strongest effect of GFP-SAUR19 overexpression can be explained by the fact that SAURs might regulate all PM H⁺-ATPases, while our genetic manipulation always influences only some of them. This corroborates the crucial role of the SAUR

proteins in mediating auxin-induced growth, and it also means that in contrast to the roots (*Pachecon-Villalobos et al., 2016*) the acid growth hypothesis seems to be valid in the hypocotyls.

## The elongation assays are relevant for the *in vivo* situation

Next we tested gravistimulation of hypocotyls to address the relevance of the results to the *in vivo* situation and to avoid any mechanical interference caused by hypocotyl decapitation and external auxin application. During gravistimulation of the shoot, auxin is redistributed to the lower side of the gravistimulated organ where it triggers growth, which in turn leads to bending of the organ (*Went and Thimann, 1937*; *Harrison and Pickard, 1989*; *Li et al., 1991*; *Friml et al., 2002*). Using the DR5rev::GFP auxin response marker (*Benková et al., 2003*) or the DII-Venus marker (*Brunoud et al., 2012*) we were unable to observe differences in auxin signaling early (1 hr) after gravistimulation. The reason is that the GFP-based markers are too slow to show differences in such a short time, and the DII-Venus marker is extremely weak in the etiolated hypocotyls, most likely due to a high level of auxin in this elongating tissue (data not shown). Gravistimulated hypocotyls bend visibly 20–40 min after gravistimulation (*Video 3*). Since our results point to acidification being an early detectable outcome of gravistimulation, we analyzed apoplastic pH in the upper and lower sides of gravistimulated etiolated hypocotyls using the apo-pHusion marker. One hour after gravistimulation, the lower side cell walls were visibly more acidic than the upper ones (*Figure 5A–C*). In non-gravistimulated hypocotyls, both sides showed either similar pH, or one of the sides was more acidic than the other, without a tendency to the left or right side (*Figure 5C*). This could reflect the fact that vertically-growing hypocotyls do not grow completely straight, but gently wave during growth (*Video 3*). Interestingly, plants expressing a stabilized version of the SAUR63 showed an increased waviness of hypocotyls and stems (*Chae et al., 2012*). *Hohm et al. (2014)* suggest that apoplast acidification is necessary for creating an auxin gradient upon directional light stimulus. Based on the fact that auxin asymmetry is established very quickly after gravistimulation (*Harrison and Pickard, 1989*), we suggest that during gravitropism, the observed apoplast acidification asymmetry is downstream of nuclear auxin signaling and is a part of the execution of auxin response. The asymmetry of acidification has been approximated before using a pH-sensitive dye to visualize differential pH in vicinity of gravistimulated aerial organs (*Mulkey et al., 1981*). We show that the apoplast itself is more acidic on the lower side of the bending hypocotyl, which fits with the results obtained with external auxin application.

The induction of the dominant-negative *axr3-1* protein completely inhibited auxin-induced growth in the elongation assay (*Figure 3B*). We therefore tested how the mutated protein affects the gravitropic response. While the heat-shocked control hypocotyls bent upwards, the *HS::axr3-1* line was utterly agravitropic (*Figure 5D*). Hypocotyl growth was almost completely inhibited as well. As expected, approximately 16 hr after the heat shock, the *HS::axr3-1* seedlings initiated bending; nicely demonstrating the transient nature of the heat-shock inducible system (*Knox et al., 2003*). This result clearly shows that the nuclear auxin response is indispensable for gravitropism, and the *in vivo* results correspond well to the elongation assay results. This result is also in agreement with the agravitropic phenotype in the stable *axr3-1* line (*Leyser et al., 1996*) and very similar to the stabilized mutant of Aux/IAA19, *msg2* (*Tatematsu et al., 2004*).

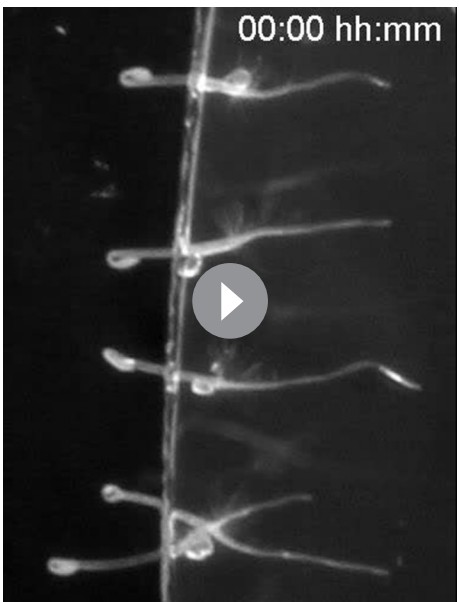

**Video 3.** Col-0 hypocotyls grown in darkness and imaged using an infrared imaging system.      At the beginning of the movie, the seedlings are stimulated by 90 degrees rotation. Seedlings bend up and continue growing upwards; note the waving of the tip of the growing seedlings. Roots are placed on agar, while the hypocotyl grows in air to allow for unobstructed bending and growth.

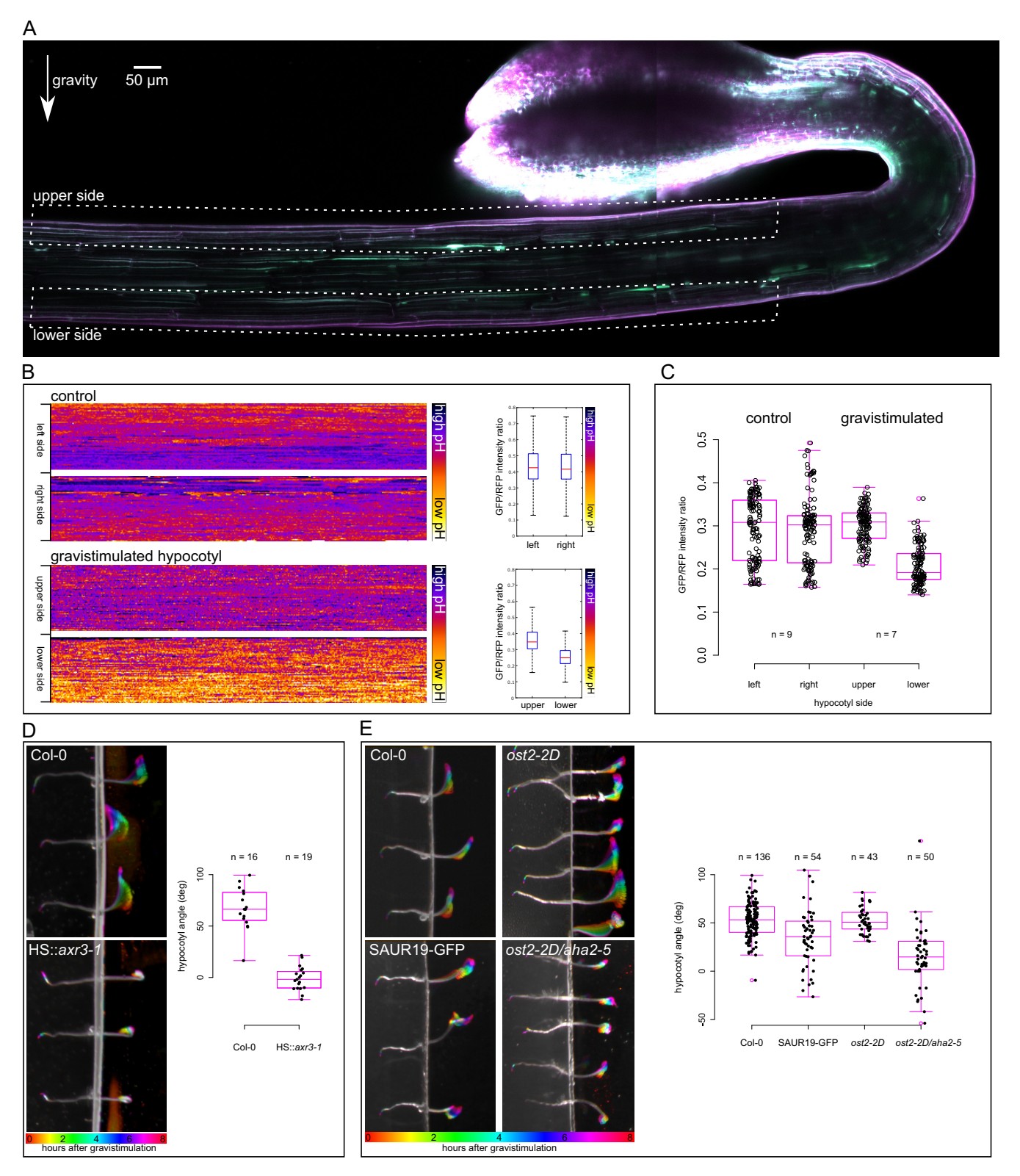

**Figure 5.** An asymmetrical acidification of the apoplast is necessary for gravitropism. (**A**) The lower epidermal side of a gravistimulated apo-pHusion marker line shows more acidic cell walls than the upper side. Hypocotyl was stimulated for 1 hr, the regions used for apoplastic pH analysis are depicted with dotted lines. SUM projection of a confocal z-stack; a 3 × 1 tilescan image. (**B**) AreaKymo representation of a control and gravistimulated hypocotyl shows that gravistimulation leads to acidification of cell walls on the lower side of the hypocotyl. The two epidermal sides cell walls are

*Figure 5 continued on next page*

*Figure 5 continued*

represented as two rectangles, and correspond to areas depicted in **A**. The quantification is shown on the right as boxplots. (**C**) Quantification of apoplastic pH in control and gravistimulated (1 hr) hypocotyls of the apo-pHusion marker line. Each point represents GFP/RFP ratio of a single z-section on either epidermal sides of hypocotyls, similar to the areas depicted in **A**. Note that the ratios cannot be directly compared to *Figures 1–3* as these images were acquired using a different microscope and sample setup (see Materials and methods for details). (**D**) Induction of the *axr3-1* protein inhibits hypocotyl gravitropism and growth. On the left, the picture shows a temporal color code of hypocotyls grown in darkness and imaged with an infrared camera for 8 hr after gravistimulation. The hypocotyl is not in contact with the agar to allow for undisturbed bending. On the right, the quantification of the hypocotyl angle 6 hr after gravistimulation is plotted; 0 degrees is horizontal, 90 degrees vertical. (**E**) The *35S::GFP-SAUR19* line is functionally insensitive to auxin in the elongation assays, and as a result, it bends less in the gravitropic assay. The constitutively active AHA1 allele in the *aha2-5* mutant background (*ost2-2D/aha2-5*) leads to diminished gravitropic response. On the right, the quantification of the hypocotyl angle 6 hr after gravistimulation is plotted; 0 degrees is horizontal, 90 degrees vertical.

The following source data is available for figure 5:

**Source data 1.** Numerical data for the graphs.

We have shown that the *35S::GFP-SAUR19* hypocotyl segments elongated in the absence of auxin, and the addition of auxin did not increase their elongation (*Figure 4A,F*). This in fact means that these hypocotyls are *functionally insensitive* to auxin or to the absence of auxin. Because of this insensitivity, this line should be unable to respond to the redistribution of auxin during gravitropism. Also the *ost2-2D/aha2-5* line showed significant auxin-independent elongation (*Figure 4B*). Therefore, we subjected the *35S::GFP-SAUR19, ost2-2D* and *ost2-2D/aha2-5* lines to gravistimulation and analyzed the bending angle after 6 hr of treatment. As shown in *Figure 5E*, lines that showed auxin-independent growth in the elongation assays also responded less to gravistimulation than the control, resulting in a smaller gravitropic bending angle. This means that the ability to respond properly to the relocation of auxin is compromised in these lines, resulting in a decreased growth gradient between the upper and lower sides of gravistimulated hypocotyls. It was shown that the *35S::GFP-SAUR19* line is less phototropic than the control (*Spartz et al., 2012*), pointing to a common role in photo- and gravitropic response. With the example of gravitropism, we showed that the results obtained in the semi *in vitro* elongation assays are relevant to the *in vivo* situation, and that apoplast acidification is one of the earliest hallmarks that can be visualized in the gravistimulated hypocotyls. The ability to rapidly induce growth after perception of auxin as well as to stop growth after withdrawal of auxin are crucial for the plants to be able to respond to directional stimuli such as gravity.

## Conclusions

In this work, we systematically re-evaluated the auxin-induced growth knowledge and test its predictions on the Arabidopsis model system using the available genetic tools. We showed that after auxin application, nuclear auxin signaling, apoplast acidification and growth occur with a similar lag phase of approximately 20 min. This is not an artifact of the model system, as the drug fusicoccin is able to trigger growth and acidification in a time substantially shorter, and without triggering the nuclear auxin response. Both the auxin-induced apoplast acidification and growth require new protein synthesis, Aux/IAA protein degradation, and are dependent on the SAUR proteins (*Spartz et al., 2014*). In other words, no non-genomic auxin pathway is required to acidify the apoplast and regulate growth in the hypocotyl. This contrasts with the situation in the root, where auxin inhibits growth very rapidly (*Monshausen et al., 2011*) and may act via different mechanism. In the hypocotyl, the *afb5-5* mutant shows a clear reduction of the reaction to the auxin analogue picloram, further confirming the involvement of the TIR1/AFB Aux/IAA auxin co-receptor complex. Furthermore, overexpression of stabilized SAUR19 protein (*Spartz et al., 2012*) causes auxin-independent hypocotyl segment growth that can be also achieved by genetic activation of PM H$^+$-ATPases, strengthening the concept of acid growth hypothesis. Using genetically-encoded apoplastic pH marker, we could directly monitor apoplast acidification *in situ* during auxin-induced growth and during gravitropism. The lines that showed auxin independent growth display problems with gravitropic bending, which demonstrates the necessity of acidification and growth gradient formation across the bending organ. It remains to be solved whether the SAUR-induced PM H$^+$-ATPases activation is the only mechanism that triggers cell elongation, or whether SAUR proteins have other targets that are responsible for

growth regulation. We are lacking tools to monitor PM hyperpolarization in-vivo, which is tightly linked to apoplast acidification, and is crucial for regulating ion fluxes over the PM and turgor regulation. It will be interesting to address the dynamics of cell wall properties and turgor pressure using this model system. The fact that auxin-induced growth of the hypocotyl utilizes the relatively slow gene transcription and protein synthesis to regulate its growth, illustrates that plant tropisms do not need to be instantaneous, as the changes in direction of gravity or light are relatively slow as well.

## Materials and methods

### Plant material and growth conditions

Col0 was used as wild type control. The marker lines and mutants originated from these publications: apo-pHusion line (*Gjetting et al., 2012*), *abp1-c1* and *abp1-TD1* (*Gao et al., 2015*), *HS::axr3-1* (*Knox et al., 2003*), *35S::GFP-SAUR19* (*Spartz et al., 2012*), *afb5-5* SALK_110643 (*Prigge et al., 2016*), ost2-2D - At2g18960 505G>T, 2599G>A (*Merlot et al., 2007*), *aha2-5* - SALK_022010, *tir1-1/ afb2-1/afb3-1* (*Dharmasiri et al., 2005*), $AHA2^{\delta95}$ (*Pachecon-Villalobos et al., 2016*). The *tir* triple and the *abp1* mutants were genotyped as described in the respective publications. The *afb5-5* was genotyped using these primers: insertion pROK-LB: GGAACCACCATCAAACAGGA + AFB5-TR2: GCATAATCTGGTTCTTGCTCACTC; wild-typeAFB5-TR2 + AFB5_F2: AAATCTTGGTGGCGTGTTG. The *ost2-2D* mutant was verified by PCR amplification of a product (ost2-2D_505F: GGA TTGGCAAGATTTTGTGGG + ost2-2D_505R: AAAACTTCTTGACCAGGGTGC) that was digested by an Hpy188III; the mutation destroys the restriction site. The *aha2-5* was genotyped by these primers: wild-type aha2-5_RP: TTGACAGGAAAGCAAACTTCTG + aha2-5_LP: ACCAAAAGGTTGTCAA TTCCATC, insertion band: aha2-5_RP + SALK_LB: ATTTTGCCGATTTCGGAAC. For most experiments, seeds were surface sterilized by chlorine gas, plated on ½ MS medium, 1% sucrose, pH 5.8, 0.8% plant agar, stratified 1–2 days at 4°C, then placed into the growthroom (21°C, long day) for ~6 hr and then wrapped into aluminum foil. The dark imaging experiments with gravitropism were done as follows: 3-d-old plants were vertically placed into dark boxes equipped with infrared LEDs (850 nm wavelength) and photographed by a Canon EOS 550D camera (as described in *Smet et al. (2014)*. The seeds were placed on the edge of agar to allow for unobstructed bending of the hypocotyls in air.

### The elongation experiment

Etiolated seedlings ~72 hr-old (counted from placing into growthroom) were decapitated using a razor blade under a dissecting scope equipped with a green filter. The segments were placed onto a cellophane foil placed on the surface of the depletion medium (10 mM KCl, 1 mM MES, pH 6 (KOH), 1.5% phytagel) and kept in darkness for 30 min. Then the segments were transferred to a new plate with depletion medium using the cellophane foil, which was removed after the transfer. Falcon 60 × 15 mm dishes were used with 5 ml of medium with the drugs or auxin treatments as indicated. The samples were placed on a flatbed scanner (Epson perfection V370) and imaged through the layer of phytagel, a wet black filter paper was placed above the open dishes to improve the contrast of the images. Samples were automatically imaged each 10 min using the AutoIt program and imaged at 2400DPI. The drug stocks were dissolved in DMSO except for fusicoccin, which was dissolved in methanol. The *HS::axr3-1* plants were heatshocked at 37°C for 40 min in aluminum-wrapped petri dishes; experiments were started 1.5 hr after the end of the heat shock. $AHA2^{\delta95}$ hypocotyl segments were transferred to depletion medium containing 1 µM dexamethasone. To image the luciferase activity, ~50 µl of 1 mM D-luciferin dissolved in PBS was added to decapitated hypocotyl segments during the depletion stage (using the depletion medium), and then transferred to the plates with treatments, and imaged. The elongation experiment, pH analysis and luciferase imaging is described in more detail at Bio-protocol (*Li et al., 2018*).

### Imaging

The hypocotyls in *Figures 1A* and *4A* were imaged using an Olympus BX53 light microscope via a 10x/0.25 PlanN objective. The apo-pHusion marker line for the elongation experiments was imaged using ZEISS LSM700 confocal microscope with a 20x/0.8 Plan-Apochromat M27 objective. GFP and RFP were imaged simultaneously, splitting the light with a SP550 and LP560 filters, 16bits per pixel,

excited using 488 and 555 nm diode lasers. Hypocotyls were decapitated and depleted on the depletion medium, as described above, then the segments were transplanted onto medium with the indicated treatment and transferred into LabTekkII chambered coverslide and imaged. Alternatively, the treatment was pipetted to the samples during imaging on the confocal microscope. For the apo-pHusion imaging during the gravitropic response, seeds were plated into channels 0.25 mm deep and 0.5 mm wide that were stamped into 1.5% agarose with ½MS, 1% sucrose. After one day of germination, the channels with seeds were covered with a microscopic coverglass and the plates were wrapped in aluminum foil and grown for additional three days. Then the plates were gravistimulated for 1 hr, and taken directly for imaging at a confocal microscope. This way, the original position of the gravistimulated organ was maintained and the handling was minimized. Zeiss LSM800 confocal microscope equipped with GaAsP detectors and using a 20x/0.8 Plan-Apochromat M27 objective was used, and imaged analogously to the ZEISS LSM700 settings. Luciferase luminescence was imaged using a darkbox equipped with a Photometrics evolve EMCCD camera with a 17 mm fixed lens/0.95. The exposure time was 110 s.

## Image analysis

All images were analyzed using Fiji (*Schneider et al., 2012*; *Schindelin et al., 2012*). The kymographs were done using the Multiple Kymograph ImageJ plugin (http://www.embl.de/eamnet/html/body_kymograph.html). To achieve unbiased measurements, the length of hypocotyl segments was measured by thresholding the scanner images, manual clicking on each segment using the Wand tool (Fiji) and recording their Feret's diameter (maximum caliper), this was done for the entire time series. The initial length of the segment was taken as 100%, data was analyzed in Excel. The results in the figures represent one of three repetitions of each experiment, which is based on quantification of 6 hypocotyl segments. For confocal time series, image drift was stabilized using the StackReg Fiji plugin in the 'translation' option. Luciferase luminescence was measured by selecting the region of interest containing the hypocotyl segment and measuring the mean intensity in this region over time. The apoplastic pH visualized by the apo-pHusion marker was measured both manually and automatically by the AreaKymo MATLAB script. Manual measurement was done by thresholding the cell wall area based on the RFP channel, using this thresholded area as a base for a region of interest (using the 'create selection' command), and then measuring the mean intensity in these regions both for GFP and RFP channels. For the gravitropic experiment, each z-section of the image was measured individually. The AreaKymo script essentially does the same procedure, but does so automatically without the user input, allowing for rapid processing of several hypocotyls at a time. The user specifies the value of the threshold for the RFP signal (this is set arbitrarily to highlight just the cell wall area) and the desired width of the rectangle that will represent the individual timeframe. The script outputs the visual representation of apoplastic pH and also the values in a form of a series of boxplots. The AreaKymo script is provided in *Supplementary file 1*. To measure the bending of hypocotyls after gravistimulation, the angle of the upper part of the hypocotyl was measured 6 hr after gravistimulation. Numerical data for the graphs and figures can be found in the Source Data files 1–6, each corresponds to *Figures 1–5* and the *Figure 3—figure supplement 1*. The hypocotyl images in *Figure 5D and E* were done using the 'temporal color code' command of Fiji, with the 'spectrum' look-up-table option. Figures were assembled in the Inkscape program (www.inkscape.org). Boxplots and stripcharts were created in R (3.2.0) via the RStudio interface.

## Acknowledgements

The authors express their gratitude to Veronika Bierbaum, Robert Hauschild for help with MATLAB, Daniel von Wangenheim for the gravitropism assay. We are thankful to Bill Gray, Mark Estelle, Michael Prigge, Ottoline Leyser, Claudia Oecking for sharing the seeds with us. We thank Katelyn Sageman-Furnas and the members of the Friml lab for critical reading of the manuscript. The research leading to these results has received funding from the People Programme (Marie Curie Actions) of the European Union's Seventh Framework Programme (FP7/2007-2013) under REA grant agreement n° 291734. This work was also supported by the European Research Council (project ERC-2011-StG-20101109-PSDP).

## Additional information

### Funding

| Funder | Grant reference number | Author |
|---|---|---|
| European Research Council | project ERC-2011-StG-20101109-PSDP | Jiří Friml |
| Marie Curie Actions FP7 2007-2013 | REA grant agreement n.291734 | Matyáš Fendrych |

The funders had no role in study design, data collection and interpretation, or the decision to submit the work for publication.

### Author contributions

MF, Conception and design, Acquisition of data, Analysis and interpretation of data, Drafting or revising the article; JL, Analysis and interpretation of data, Drafting or revising the article, Contributed unpublished essential data or reagents; JF, Conception and design, Drafting or revising the article

### Author ORCIDs

Jiří Friml, http://orcid.org/0000-0002-8302-7596

## Additional files

### Supplementary files

• Supplementary file 1. The AreaKymo script.

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
