## [Decision Letter]

Thank you for submitting your article "TIR1/AFB-Aux/IAA auxin perception mediates rapid cell wall acidification and growth of Arabidopsis hypocotyls" for consideration by *eLife*. Your article has been favorably evaluated by Christian Hardtke (Senior Editor) and three reviewers, one of whom is a member of our Board of Reviewing Editors. The reviewers have opted to remain anonymous.

The reviewers have discussed the reviews with one another and the Reviewing Editor has drafted this decision to help you prepare a revised submission.

Please prepare your revisions carefully and address the reviewers' demands one by one, particularly with respect to the existing, and most recent literature on the topic.

Summary:

This manuscript concerns the function of the plant hormone auxin in rapid elongation of the Arabidopsis hypocotyl. Evidence accumulated over many years suggests that cell expansion in the hypocotyl occurs through acid growth. Auxin is thought to activate plasma membrane H-ATPase, either indirectly or indirectly, and thereby acidify the apoplast resulting in changes in the cell wall. Recent studies suggest that the H-ATPase is activated by phosphorylation, and that this achieved by SAUR dependent inhibition of a PP2C phosphatase. Although there is some evidence for each component of this model, there is no information of the auxin receptor that mediates this growth response. In this paper, the authors use excised hypocotyls from etiolated seedlings as their model. Using this system, they show that changes in growth and acidification of the apoplast are correlated in time. Further, they show that these responses are dependent on protein synthesis implying that they involve changes in transcription/translation. A novel and important finding is that growth is also correlated with an increase in auxin transcriptional response as measured by a DR5:LUC reporter. Most significantly they also show that these effects are dependent on the TIR1/AFB receptor system but not ABP1. The authors make elegant use of the afb5-5 mutation. These plants are resistant to the synthetic auxin picloram but are otherwise normal in appearance. By showing that afb5 is resistant to the rapid effects of picloram on growth, the authors demonstrate that AFB5 is required for the response to auxin. It follows that the TIR1/AFB receptors act to mediate the response to endogenous auxin.

Critique and required revisions:

All of the reviewers agreed that the problem being addressed by the manuscript is an important one and they all appreciated the care with which experiments have been done. However, there were concerns about the overall novelty of the work, issues about organization and presentation, as well as some specifics related to certain experiments. I have outlined these below taking, whenever possible, text directly from the various reviews.

1) The reviewers felt that the real novelty in the paper had to do with the results using AFB5. Much of the rest is technically excellent but largely confirms previous work. Hence, the authors need to clarify in the Introduction, as well as the Discussion, which aspects of the current work truly are novel and which are confirmatory to previous findings. Here is just one example of the concerns that the reviewers raised in this regard:

"The last section of the manuscript describes an attempt to validate these new results in vivo by examining the effects of axr3-1, GFP-SAUR19 on gravitropism. […] The technology used is more sophisticated in this study, but the bottom line is the same."

Similar comments were made with regard to other sections of the manuscript so there is a real need to clarify for the reader what aspects of the paper are truly novel.

2) A very important question is, 'is there a RAPID response to auxin in Arabidopsis that is related to auxin signaling?' The authors show that in contrast to fusicoccin, which causes growth and apoplast acidification within seconds and minutes, auxin requires 15-20 minutes for an effect. These results are in complete agreement with our current model where fusicoccin is binding to and stabilized the activated form of the pump (via binding to the phopshoTh947 in concert with 14-3-3) whereas the SAUR gene must be expressed as a mRNA and then translated to make a SAUR protein that then binds to the protein phosphatase that dephosphorylates Thr947. There is a very big difference in time dependence for growth and apoplastic pH changes between auxin and FC and this is a critical important observation. The authors should explicitly state that the earliest auxin response they see is 15 minutes or more that is not an artifact of their measurement because fusicoccin causes a rapid immediate response. The authors ultimate conclude that they see no such rapid response with auxin but this is only mentioned briefly in the discussion. This is an example of how the writing of the paper obscures some of its significance. The authors should clearly state in the introduction the key questions being addressed and then emphasize in a clear, concise way in the discussion how their data relates to these questions.

3) In the Abstract, the acid growth hypothesis is presented as established fact. Although the model is attractive, the importance of the H-ATPases to auxin induced growth seems clear but not airtight. The aha1 and aha2 KO mutants have very subtle growth related phenotypes although a double is lethal. The proposed function of the SAURs is based mainly on stabilized gain-of-function fusion proteins. In other words, we know that increased levels of GFP-SAUR19 are sufficient to promote growth, but we don't yet know if the SAURs are all that is necessary for the 15 minute auxin induced growth. This uncertainty should be reflected in this manuscript. It may be worth checking the rapid fusicoccin response in all the mutants and also, in the presence of auxin, to make sure that auxin or the TIR mediated auxin signaling pathway is not doing something unusual to prevent the rapid response (e.g., by FC) under the conditions of these experiments.

4) In the Introduction the authors state "Despite these recent findings, there is also ample evidence demonstrating the importance of the ABP1 pathway". This statement and those that directly follow should be deleted or modified. The lack of phenotype of the abp1 null mutant clearly indicates that ABP1 is not important in Arabidopsis under any conditions so far reported. The physiological significance of protoplast swelling is unknown and the phenotype of the tmk mutants clearly has nothing to do with ABP1 since the abp1 mutant does not have this phenotype.

5) At several points the authors use the word "perfect" to describe something. Since nothing is perfect, words like "excellent" or "outstanding" would be better.

6) Describing an experimental system as "spectacular" (subsection “Stabilization of SAUR19 protein leads to auxin-independent growth”, first paragraph) is a bit over the top. How about "remarkable"?

7) In the first paragraph of the subsection “Stabilization of SAUR19 protein leads to auxin-independent growth”, "nearly instantaneous" is a bit strong. Presumably several events must occur before transcription can begin, like histone acetylation, chromatin remodeling etc. How about “very rapidly".

8) In the subsection “Conclusions” the authors state: "We also show that auxin-induced apoplast acidification is dependent on nuclear auxin signaling and is mediated via the SAUR proteins" Surely the role of the SAURs has already been demonstrated by the Gray lab (Spartz et al.). Again, refer to point one above.

9) It is important for these authors to consider the possibility that all of the rapid-auxin electrophysiological responses are in fact related to transport of auxin rather than action. Since auxin uptake is likely to involve proton mediated symporters or antiporters and auxin efflux is likely to be an electrogenic efflux of the negatively charged auxin ion, there are likely to be very rapid changes in the PMF (equal to sum of δ pH and EMF) that may be small and unrelated to the auxin growth response. Use of citations in the literature (e.g. protoplast swelling) that are not replicated in this work, and whose reproducibility and relevance to auxin induced growth as measured in this carefully done report, detracts from its credibility to readers. For example, in the subsection “Conclusions”, it is unclear why the protoplast swelling phenomenon is even mentioned since it has nothing to do with this paper. On face value the lack of a phenotype in the abp1 mutant suggests that the effects of ABP1 on protoplasts are not important. Given the importance of auxin action in plant life, and the confusion and poorly interpreted experiments presented in the past, the authors should pause, stay close to their own data and double check their writing and interpretations.

10) The difference between afb5-5 and the other tir1/afb lines may be due to specialized function of AFB5. The authors may want to note this.

11) In Figure 1A, a little bit more information to help the reader understand the kymograph would be useful. Perhaps an arrow indicating time. It also might not be obvious that the bar refers to 10 min. In this regard, it looks like much more growth occurs in the apical part of the segment than in the basal part contrary to what is stated in the figure legend.

12) In the first paragraph of the subsection “Auxin-induced growth requires TIR1/AFB-Aux/IAA signaling”: change to 'most likely due to the redundancy[…]". The authors take the evidence to unequivocally conclude that it is gene redundancy that explains their results but such a conclusion is not justified by the data presented.

---

## [Author Response]

*1) The reviewers felt that the real novelty in the paper had to do with the results using AFB5. Much of the rest is technically excellent but largely confirms previous work. Hence, the authors need to clarify in the Introduction, as well as the Discussion, which aspects of the current work truly are novel and which are confirmatory to previous findings. Here is just one example of the concerns that the reviewers raised in this regard:*

*"The last section of the manuscript describes an attempt to validate these new results in vivo by examining the effects of axr3-1, GFP-SAUR19 on gravitropism. […] The technology used is more sophisticated in this study, but the bottom line is the same."*

*Similar comments were made with regard to other sections of the manuscript so there is a real need to clarify for the reader what aspects of the paper are truly novel.*

In the re-submitted version, we attempted to clarify the issue of novelty more, and we tried to be more explicit and less ‘obscure’. We rewrote the end of the Introduction and the conclusion section completely as well.

We propose that the true importance of our paper is bringing the old and new knowledge about auxin-induced growth into one experimental system, and systematically focus on and analyze this phenomenon. More specifically, we clarified how auxin is perceived during this process, and showed that there are no obvious ‘rapid’ responses to auxin in the hypocotyl growth. Very importantly, by visualizing apoplastic pH we also demonstrate that auxin-induced acidification is not a ‘rapid, non-genomic’ effect, but instead requires the nuclear auxin signaling pathway. These questions were unanswered in the field and needed to be sorted out. We did not want to specifically state which pieces are novel in our text, we think this is not appropriate, but here we can state that not only the above mentioned questions have really not been answered, but also the apoplastic pH in situ observation after auxin application has not been shown before, as wasn’t the apoplastic pH behavior during gravitropism. Yes, the behavior of stabilized Aux/IAA mutant during gravitropism has been shown before, but we also show that the ost2-2D/aha2-5 mutant has a problem bending, which we put into context of the elongation assays and compare it to the behavior of the GFP-SAUR19 plants. Because of the nature of the manuscript, we were trying to be extremely careful with referencing the existing literature wherever possible throughout the text. We would welcome any specific suggestion on literature that we might have missed. In summary, this work clarified several decades-lasting controversies and given how important the auxin-induced growth is as experimental topic in plant hormone biology – we find this a very important contribution.

*2) A very important question is, 'is there a RAPID response to auxin in Arabidopsis that is related to auxin signaling?' The authors show that in contrast to fusicoccin, which causes growth and apoplast acidification within seconds and minutes, auxin requires 15-20 minutes for an effect. These results are in complete agreement with our current model where fusicoccin is binding to and stabilized the activated form of the pump (via binding to the phopshoTh947 in concert with 14-3-3) whereas the SAUR gene must be expressed as a mRNA and then translated to make a SAUR protein that then binds to the protein phosphatase that dephosphorylates Thr947. There is a very big difference in time dependence for growth and apoplastic pH changes between auxin and FC and this is a critical important observation. The authors should explicitly state that the earliest auxin response they see is 15 minutes or more that is not an artifact of their measurement because fusicoccin causes a rapid immediate response. The authors ultimate conclude that they see no such rapid response with auxin but this is only mentioned briefly in the discussion. This is an example of how the writing of the paper obscures some of its significance. The authors should clearly state in the introduction the key questions being addressed and then emphasize in a clear, concise way in the discussion how their data relates to these questions.*

We entirely agree with this point. There is an important argument to think that the delay of auxin action is not an artifact of the system – the behavior of the DR5::LUC reporter. The earliest observable LUC signal appears 20 minutes after auxin application, similarly to acidification and growth onset. The DR5 promoter needs to be activated, luciferase protein needs to be transcribed and translated and the luminescent product needs to accumulate to be detected. Therefore, it is probable that auxin enters the hypocotyl very rapidly, and the delay is caused by the auxin signaling machinery and transcription and translation of the executor genes.

To test whether FC can indeed act ‘rapidly’, we performed additional experiments focusing on its early action on hypocotyl elongation and triggering apoplast acidification. These data are now included as Figure 1 and Figure 2 and Video 2 (we split the Figure 1 into two figures to accommodate the data). After application of 5uM fusicoccin, the first detectable elongation occurred in 6-8 minutes. Also the acidification of the apoplast was apparent immediately after application of FC. We rewrote the end of the Introduction to clarify the questions we are trying to answer, and we rewrote the conclusion section as well.

*3) In the Abstract, the acid growth hypothesis is presented as established fact. Although the model is attractive, the importance of the H-ATPases to auxin induced growth seems clear but not airtight. The aha1 and aha2 KO mutants have very subtle growth related phenotypes although a double is lethal. The proposed function of the SAURs is based mainly on stabilized gain-of-function fusion proteins. In other words, we know that increased levels of SAUR19-GFP are sufficient to promote growth, but we don't yet know if the SAURs are all that is necessary for the 15 minute auxin induced growth. This uncertainty should be reflected in this manuscript. It may be worth checking the rapid fusicoccin response in all the mutants and also, in the presence of auxin, to make sure that auxin or the TIR mediated auxin signaling pathway is not doing something unusual to prevent the rapid response (e.g., by FC) under the conditions of these experiments.*

We agree that the interpretation of the SAURs role, which is based on overexpression of stabilized protein, is an important point that was missing in our interpretation and we included this in the end of the respective paragraph.

However, we do not agree that we presented the acid growth hypothesis as a fact. We show in the previous version of the paper that the acid growth hypothesis might be more complicated than just activation of H-ATPases. We show that the overexpression of SAURs has much stronger effect than the presence of a constitutively active AHA1 form (the ost2-2D mutant), where the auxin-independent growth was not observed. We could see auxin-independent growth when the active AHA1 (ost2-2D) was in the aha2-5 mutant background, but still the magnitude of the response could not be compared to the effect of SAUR19-GFP overexpressor. During the review process, the Pacheco-Villalobos et al. paper introduced the AHA2^delta95^ line, where the non-regulated, active AHA2 ATPase can be expressed inducibly. We tested this line for the auxin-independent growth, and found that ~4 hours after the transgene induction, a rapid growth can be observed. Even though this again is an experiment based on the strong overexpression, it clearly shows that activation of H-ATPases is sufficient to trigger elongation of the hypocotyl segments. Still, acid growth hypothesis is probably a simplification, as the acidification of the apoplast is only a part of the processes – activation of the pump will hyperpolarize the membrane and increase the proton motive force, which is crucial for maintaining or increasing the turgor pressure. We added this data in Figure 4, and added an extra paragraph to the text.

In the previous version of the manuscript, we showed that the HS::axr3-1 line unable to respond to auxin still responded normally to FC. In the new version, we included the fusicoccin growth data for afb5-5, SAUR19-GFP, ost2-2D and ost2-2Dxaha2-5 mutants as well; see Figure 4 and the text.

*4) In the Introduction the authors state "Despite these recent findings, there is also ample evidence demonstrating the importance of the ABP1 pathway". This statement and those that directly follow should be deleted or modified. The lack of phenotype of the abp1 null mutant clearly indicates that ABP1 is not important in Arabidopsis under any conditions so far reported. The physiological significance of protoplast swelling is unknown and the phenotype of the tmk mutants clearly has nothing to do with ABP1 since the abp1 mutant does not have this phenotype.*

We modified the text. For the protoplast swelling, please see the point 9 of this response.

*5) At several points the authors use the word "perfect" to describe something. Since nothing is perfect, words like "excellent" or "outstanding" would be better.* We changed ‘perfect’ as suggested.

*6) Describing an experimental system as "spectacular" (subsection “Stabilization of SAUR19 protein leads to auxin-independent growth”, first paragraph) is a bit over the top. How about "remarkable"?* We changed ‘spectacular’ to ‘remarkable’.

*7) In the first paragraph of the subsection “Stabilization of SAUR19 protein leads to auxin-independent growth”, "nearly instantaneous" is a bit strong. Presumably several events must occur before transcription can begin, like histone acetylation, chromatin remodeling etc. How about “very rapidly".*

We changed the text as suggested.

*8) In the subsection “Conclusions” the authors state: "We also show that auxin-induced apoplast acidification is dependent on nuclear auxin signaling and is mediated via the SAUR proteins" Surely the role of the SAURs has already been demonstrated by the Gray lab (Spartz et al.). Again, refer to point one above.*

Here the main point of the sentence was supposed to be that apoplast acidification needs nuclear auxin signaling –i.e. there are no rapid pathways that acidify the apoplast. We rewrote the sentence as follows: “Both the auxin-induced apoplast acidification and growth require new protein synthesis, Aux/IAA protein degradation, and are dependent on the SAUR proteins (Spartz et al., 2014). In other words, no non-genomic auxin pathway is required to acidify the apoplast and regulates growth in the hypocotyl.”We hope it is clearer now?

*9) It is important for these authors to consider the possibility that all of the rapid-auxin electrophysiological responses are in fact related to transport of auxin rather than action. Since auxin uptake is likely to involve proton mediated symporters or antiporters and auxin efflux is likely to be an electrogenic efflux of the negatively charged auxin ion, there are likely to be very rapid changes in the PMF (equal to sum of δ pH and EMF) that may be small and unrelated to the auxin growth response. Use of citations in the literature (e.g. protoplast swelling) that are not replicated in this work, and whose reproducibility and relevance to auxin induced growth as measured in this carefully done report, detracts from its credibility to readers. For example, in the subsection “Conclusions”, it is unclear why the protoplast swelling phenomenon is even mentioned since it has nothing to do with this paper. On face value the lack of a phenotype in the abp1 mutant suggests that the effects of ABP1 on protoplasts are not important. Given the importance of auxin action in plant life, and the confusion and poorly interpreted experiments presented in the past, the authors should pause, stay close to their own data and double check their writing and interpretations.*

We thought protoplast swelling was a clear example of shoot cells that show a truly ‘rapid’ response to auxin; something we do not see in the case of acidification and growth of the hypocotyl. That was the reason to include protoplast swelling references. The possibility that the rapid electrophysiological effects of auxin are actually connected with auxin transport is interesting. We do not feel competent to say that the protoplast swelling papers that we referred to are ‘poorly interpreted’. Anyway, we removed the protoplast swelling from the paper conclusions, because indeed it was inappropriate there, but we kept it in the Introduction, for the above mentioned reasons.

*10) The difference between afb5-5 and the other tir1/afb lines may be due to specialized function of AFB5. The authors may want to note this.*

That is a good point, we included it into the text.

11) In Figure 1, a little bit more information to help the reader understand the kymograph would be useful. Perhaps an arrow indicating time. It also might not be obvious that the bar refers to 10 min. In this regard, it looks like much more growth occurs in the apical part of the segment than in the basal part contrary to what is stated in the figure legend.

We changed theFigure 1 and Figure 4, as suggested. We added an arrow of 20-min length and arrows indicating which dimension is space and which time. The bottom part of the segment is fixed and so in the kymograph, the growth progressively sums up towards the apical part, but growth can also be seen in the lower third of the segment.

12) In the first paragraph of the subsection “Auxin-induced growth requires TIR1/AFB-Aux/IAA signaling”: change to 'most likely due to the redundancy[…]". The authors take the evidence to unequivocally conclude that it is gene redundancy that explains their results but such a conclusion is not justified by the data presented.

We changed the text as suggested.